# Dynamics of local productive arrangements in the municipalities of Mato Grosso do Sul considering the transformations of the Bioceanic Corridor

**Mateus Boldrine Abrita**[1] *, **Rafaella Stradiotto Vignandi**[2], **Daniel Amorim Souza Centurião**[1], **Angelo Rondina Neto**[3], **Ana Paula Camilo Pereira**[1], **Guilherme Espindola Junior**[4], **Nelagley Marques**[1], **Vanessa Aparecida de Moraes Weber**[1,5], **Ruberval Franco Maciel**[1]

**1** State University of Mato Grosso do Sul, Campo Grande, Mato Grosso do Sul, Brazil, **2** Federal University of Rondonópolis, Rondonópolis, Mato Grosso, Brazil, **3** State University of Londrina, Londrina, Paraná, Brazil, **4** Municipal Environment and Urban Planning Agency, Campo Grande, Mato Grosso do Sul, Brazil, **5** KeroW Soluções de Precisão, Campo Grande, Mato Grosso do Sul, Brazil

\* mateusabrita@uems.br

**Data Availability Statement:** The data used in the research were obtained from public primary sources of the Brazilian federal government, publicly available online. These refer to information

## Abstract

The Bioceanic Corridor is an international land route under implementation, which aims to connect the State of Mato Grosso do Sul, in Brazil, to the ports of Northern Chile. This new route could shorten the transport time between South America and Asia by approximately two weeks. This paper's purpose is to contextualize, map, identify, and analyze the effects of this new logistics network designed by the Bioceanic Route over the Local Productive Arrangements (LPA) in the State of Mato Grosso do Sul. To achieve these goals, a spatial econometric methodology was adopted to determine the State's productive concentration. The results indicate that this route will bring many development opportunities. However, for that to happen, favorable policies are essential and must be developed to facilitate an integration that allows for competitiveness in the State's economic activities. However, mere unplanned integration will potentially only lead to the reinforcement of already existing regional inequalities in the State.

## Introduction

Bioceanic Corridor (CB), Bioceanic Route (RB), or *Rota de Integração Latino Americana* (RILA) is an international road transport corridor that is under implementation; it aims to connect the State of Mato Grosso do Sul to the ports of Northern Chile for the flow of goods. The name "bioceanic" comes from the possibility of connecting Brazilian ports in the Atlantic Ocean to the ports of northern Chile in the Pacific Ocean. In this context, the general objective of this article is to contextualize, map, identify, and analyze the productive arrangements in Mato Grosso do Sul, from the new logistics network designed by the Bioceanic Route. Thus, the main objectives were to 1) identify and characterize the municipalities of Mato Grosso do

from the Annual Social Information Report (RAIS), a self-declaration that all companies (legal entities) must submit to the federal government (Ministry of Labor) at the end of each year. Information on the primary database can be found at the link: http://pdet.mte.gov.br/rais. Specifically, the microdata are also available via an FTP server, via the link: ftp://ftp.mtps.gov.br/pdet/microdados/. Another possibility of consultation would be through the BI provided by the Ministry of Labor, available at: https://bi.mte.gov.br/bgcaged/login.php. In this space, any user can use the login, "basico", and password, "12345678", to access (such login and password information are public available by the Ministry of Labor of Brazil). With the primary public information data, we filtered the information for the spatial and temporal subset that we used in the research and performed the calculation of the spatial indices. Such a database elaborated in our research is available in the Mendeley database repository, available for public access at the link: https://data.mendeley.com/datasets/bjf5f5zkwh, DOI:10.17632/bjf5f5zkwh.2

**Funding:** all authors 12 Grants SUDECO - Superintendência do Desenvolvimento do Centro-Oeste https://www.gov.br/sudeco/pt-br The funders had no role in study design, data collection and analysis, decision to publish, or preparation of the manuscript.

**Competing interests:** The authors have declared that no competing interests exist.

Sul participating in the Bioceanic Corridor; 2) analyze the productive structure of municipalities through specialization indicators; and 3) determine the indicators of productive specialization of selected municipalities.

For this, a spatial econometry methodology was adopted for the analysis of productive concentration. The main contributions of this article include i) route mapping; ii) spatial identification and analysis of the productive structure of the municipalities of Mato Grosso do Sul participating in the bioceanic corridor; iii) the analysis of the characteristics of the productive specialization of the selected municipalities via concentration indicators; and iv) the mapping of local productive arrangements in Mato Grosso do Sul. These contributions are innovative and unprecedented, as this transport route is still under construction and hasn't been under the scrutiny of robust academic literature that develops these points yet.

In view of the foregoing, this article is organized into four sections, in addition to the Introduction and the Final Considerations. Initially, a contextualization of the Bioceanic Corridor project is presented, including the maps of its route and the site of the bridge as well as the cornerstone of the bioceanic bridge construction. Next, an analysis is developed on the location, identification, and economic characterization of the municipalities of Mato Grosso do Sul in relation to the Bioceanic Corridor. In the third section, the methodology, as well as the data used in the econometric analysis, are presented. In the fourth section, the analysis of the locational quotient and indicators of productive specialization are performed. Finally, considerations are presented with some of the main conclusions.

## The Bioceanic Corridor

Bioceanic Corridor (CB), Bioceanic Route (RB), or *Rota de Integração Latino Americana* (RILA) is an international road transport corridor under implementation that aims to connect the State of Mato Grosso do Sul to the ports of Northern Chile for the flow of goods. The name "bioceanic" comes from the possibility of connecting Brazilian ports in the Atlantic Ocean to the ports of northern Chile in the Pacific Ocean, and "RILA" emphasizes the possibility of strengthening the integration of people in Latin America, especially as it is a corridor made possible mainly by road. Regardless of the name, this project has generated important debates in society, especially because of the enormous potential it offers for transformations. Thus, it is essential to advance research that has this undertaking as the object of study. In this article, these three different forms of nomenclature will be adopted to broadly achieve the support of readers.

The possibility of a new goods transport route connecting the Port of Santos, in the State of São Paulo, Brazil, to the ports of Antofagasta and Iquique in Chile, passing through the State of Mato Grosso do Sul (MS), Paraguay, and Argentina, has opened up intense debates about the possible economic and social transformations and opportunities, especially in the field of regional development.

According to [1], this transport corridor will result in various gains, such as a reduction in terms of cost and time in transportation, storage, and inventory, a promotion in the movement of cargo and passengers, an encouragement for the formation of strategic, productive partnerships and the development of productive integration projects, and value aggregation in the countries of origin and destination, in addition to the en-route countries. Indeed [2], made important notes on a meeting at the end of 2015 in Asunción, Paraguay. On this occasion, several countries sent their representatives, including the Heads of State of the Southern Common Market (MERCOSUR). This meeting resulted in the creation of projects to conduct technical studies and assist in the construction of this corridor, amongst others.

Considering that the corridor construction project is underway, with the aim to seek an initial socioeconomic understanding [3], performed a characterization through the economic indicators of municipalities of Mato Grosso do Sul (MS), such as the Gross Domestic Product (GDP), GDP *per capita*, population, tax collection, gross sector value added, and main, secondary, and tertiary economic activities.

After the analyses of these indicators [3], pointed out that the municipality of Porto Murtinho is the most vulnerable from an economic and social point-of-view, with low productive dynamics and the economy being mainly based on livestock and public administration resources, despite the growth of the tourism sector. In this sense, the authors highlight the importance of public and private promotion policies in maximizing the possibility of beneficial results for the entire State of Mato Grosso do Sul. In this context of economic and social challenges—especially for the State of Mato Grosso do Sul, in which it is necessary to foster the positive potential and minimize the negative impacts—the article has as its main objective the analysis of the territorial productive structure of the important municipalities of Mato Grosso do Sul that will be affected by the RILA route.

When we conduct a historic review, we observe that the longing to integrate the Andean peoples, as well as South America, is not new. According to [4], ever since the creation of Mercosur, there has been an interest in physically connecting countries, by connecting the Pacific Ocean to the Atlantic Ocean. In this respect, some important cities will be directly influenced by this corridor, such as Campo Grande and Porto Murtinho in the State of Mato Grosso do Sul in Brazil; Carmelo Peralta, Mariscal José Félix Estigarribia, Boquerón, and Pozo Hondo in Paraguay; Misión La Paz, Tartagal, Jujuy, and Salta in Argentina; and Mejillones and Iquique in Chile. It is also important to highlight the Port of Antofagasta in Chile.

Regarding the cultural aspects [4], points out that Brazil, Paraguay, and Argentina share some common cultural elements pertaining to their gastronomic habits, religion, dances, folk performances, and some indigenous heritage. Thus, in addition to providing economic integration, RILA can generate cultural exchange.

Also, according to [4, 5], the process of road interconnection in the Americas had an important initiative on the part of the South American Council on Infrastructure and Planning (Cosiplan). This work front aimed to develop the initiative on the Integration of South American Regional Infrastructure (IIRSA) to foster more efficient processes related to infrastructure, transport, and logistics in South America. IIRSA is a multinational, multisectoral, and multidisciplinary initiative involving countries in South America in which transport, energy, and telecommunications sectors participate and where the economic, legal, political, social, cultural, and environmental aspects are involved [6]. A relevant milestone occurred in August 2000, in the city of Brasilia, Brazil, with the meeting of the Presidents of South American countries. On that date, an integration and cooperation plan was formed with a focus on the Integration and Development Axes (EID) of South America, which again underwent some changes in 2004. An EID is a multinational strip of land that includes a certain allocation of natural resources, human settlements, productive areas, and coordination services. This area is articulated by transport, energy, and communications infrastructure that facilitates the flow of goods and services, people, and information both within its own territory as well as to and from the rest of the world [7].

According to [8], the bioceanic route will be approximately 2200 kilometers long, supposing it starts in the city of Campo Grande (MS) and ends in northern Chile, where there are ports for the flow of goods. An important milestone for this project was the expedition, conducted by entrepreneurs and representatives of public institutions, named RILA, in the State of Mato Grosso do Sul. One of the objectives of this caravan was to observe the most feasible route for the implementation of this international road corridor. This expedition took place

from August 25 to September 2, 2017. Thus, according to [8], one of the most feasible routes was the one pointed out in Fig 1.

Thus, to enable RILA, it would be necessary to build a bridge connecting Porto Murtinho (2) to the municipality of Carmelo Peralta (3), the border between Brazil and Paraguay, according to points numbered in Fig 1. The complete map of the route shows the road passing through 12 cities along the entire corridor. The first international treaty ever signed concerns the construction of the bridge connecting Porto Murtinho (2), in Brazil, to Carmelo Peralta (3), in Paraguay.

Recently, there has been considerable development in the process of building infrastructure to make the route viable. According to the official data from the municipality of [10], asphalting in Paraguay advanced substantially and, on February 25, 2022, approximately 275 kilometers of pavement was inaugurated on the stretch between Loma Plata and Carmelo Peralta in Paraguay, thereby completing an important stage in the implementation of the road construction of the Bioceanic Route. The cornerstone of the construction of the international bridge over the Paraguay River, between Porto Murtinho and the Paraguayan city of Carmelo Peralta, was planned for December 13, 2021 [11]. However, according to the municipality of [12], the event was postponed to the year 2022 due to the impossibility of the Brazilian presidential helicopter to land in the region due to its weather conditions. The site where the bridge will be built can be seen in Fig 2 below.

In accordance with the *República Federativa do Brasil e República do Paraguai* [14], the construction of the bridge is expected to be completed in 2023; being the investment of around 75 million dollars and according to predictions, it will be 680 meters long, containing 380 meters of free spans and 22 meters in height, with two towers over 100 meters in height and viaducts of 150 meters at both headboards, with pillars that allow the circulation of large vehicles. More than a thousand jobs are expected to be created from this. Therefore, the infrastructure to make this international road corridor viable is under construction and should be completed in the coming years. However, due to the Covid-19 pandemic, this deadline is likely to be postponed.

Moving on from the brief introduction to the RILA, the aspects related to the identification, location, and socioeconomic characteristics of the municipalities of Mato Grosso do Sul will be developed and analyzed, especially for those municipalities situated along the corridor path.

## Mato Grosso do Sul and the bioceanic corridor: location, identification, and economic characterization

The RILA project has generated recurrent socioeconomic and scientific debates, especially in Mato Grosso do Sul and in the Midwest region of Brazil, either due to the reflections on material and immaterial circulations or due to economic development and/or the consequential territorial planning [15].

As highlighted in the previous section, this road path according to [16], has the purpose of connecting the countries of Brazil, Paraguay, and Argentina to the ports of northern Chile. These ports are relevant, especially in the cities of Iquique, Antofagasta, and Mejillones, because they have competitive port tariffs and allow easier access to Asia and its geological structure, facilitating access to larger ships. The viability of this corridor is projected from the capital, Campo Grande, the capital of Mato Grosso do Sul through the bordering municipality of Porto Murtinho, via the north of Paraguay and Argentina, toward the ports of Northern Chile, according to Fig 3 below.

The route path may provide socioeconomic benefits, such as 1) reducing the transit time and the cost of transport, storage, and inventory; 2) stimulating the use of more than one

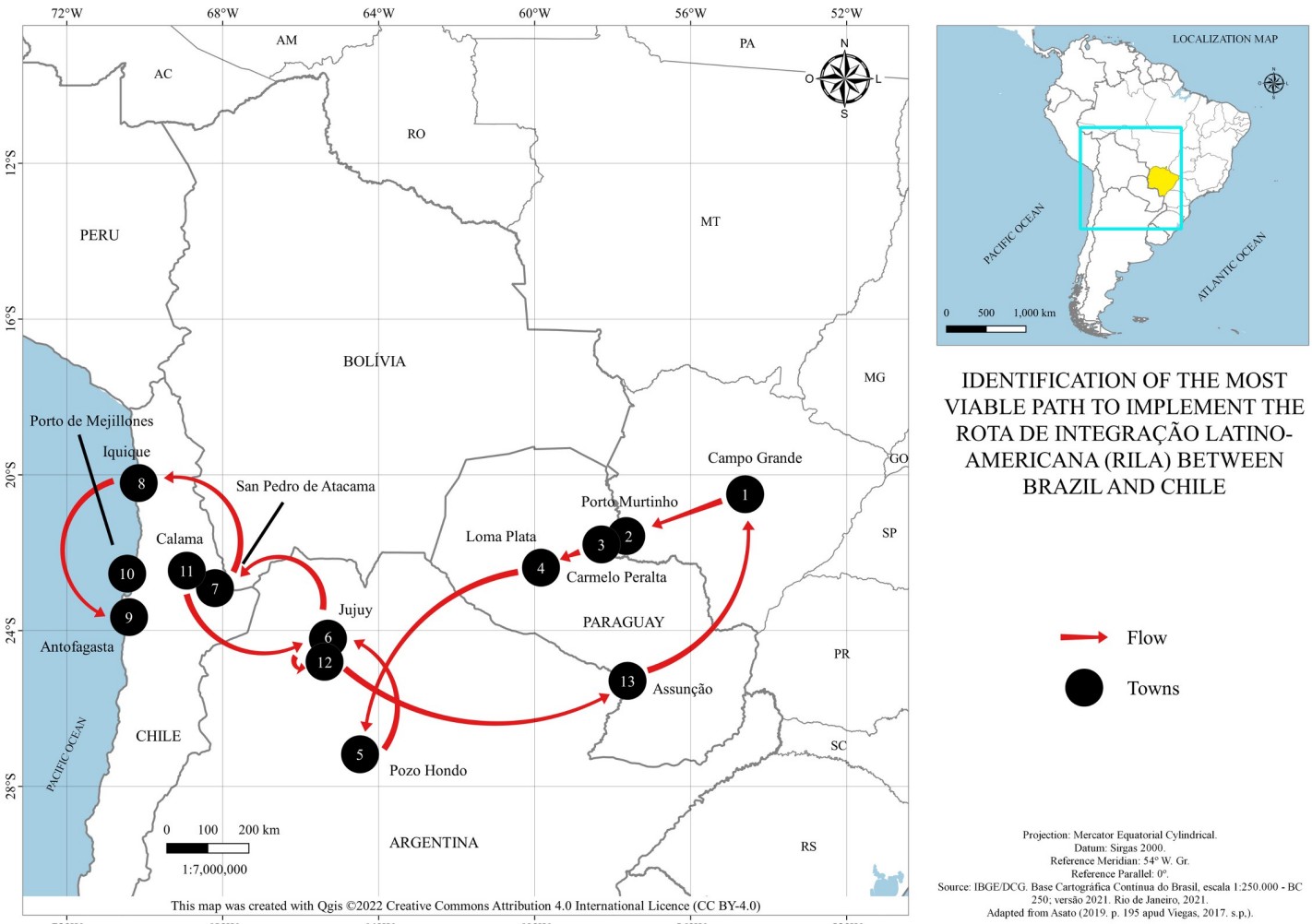

**Fig 1. The identification of the most viable path for implementing the *Rota de Integração Latino-Americana* (RILA) between Brazil and Chile.** Caption credit: Own development based on data from [9]. Adapted from [8].

modal; 3) generating efficient cargo and passenger transport in terms of reliability, predictability, and safety; 4) stimulating partnerships; and 5) furthering the development of productive integration projects and the aggregation of value in countries of origin and of destination, as well as in transit countries [1, 16], for promoting a direct impact on the circulation, and consequently, on spatial planning. The following Fig 4 shows the geographical identification of the municipalities of Mato Grosso do Sul that are impacted directly and indirectly by the corridor.

In blue, we have the municipalities of Mato Grosso do Sul such that even though they are not on the specific route—that is, their municipal headquarters are not in the outline—the territorial boundary of the municipality is in the path. In yellow, the specific municipalities in which their cities are in the path of the Bioceanic Route are shown.

Considering this scenario, from Campo Grande, we have geographically situated in sequence the municipalities of Sidrolândia, Nioaque, Guia Lopes da Laguna, Jardim, and Porto Murtinho, all of which are in the State of Mato Grosso do Sul and, after Porto Murtinho, the route runs through Paraguay, Argentina, and Chile, respectively. When considering the Bioceanic Corridor from the Port of Santos in the State of São Paulo, in this case connecting

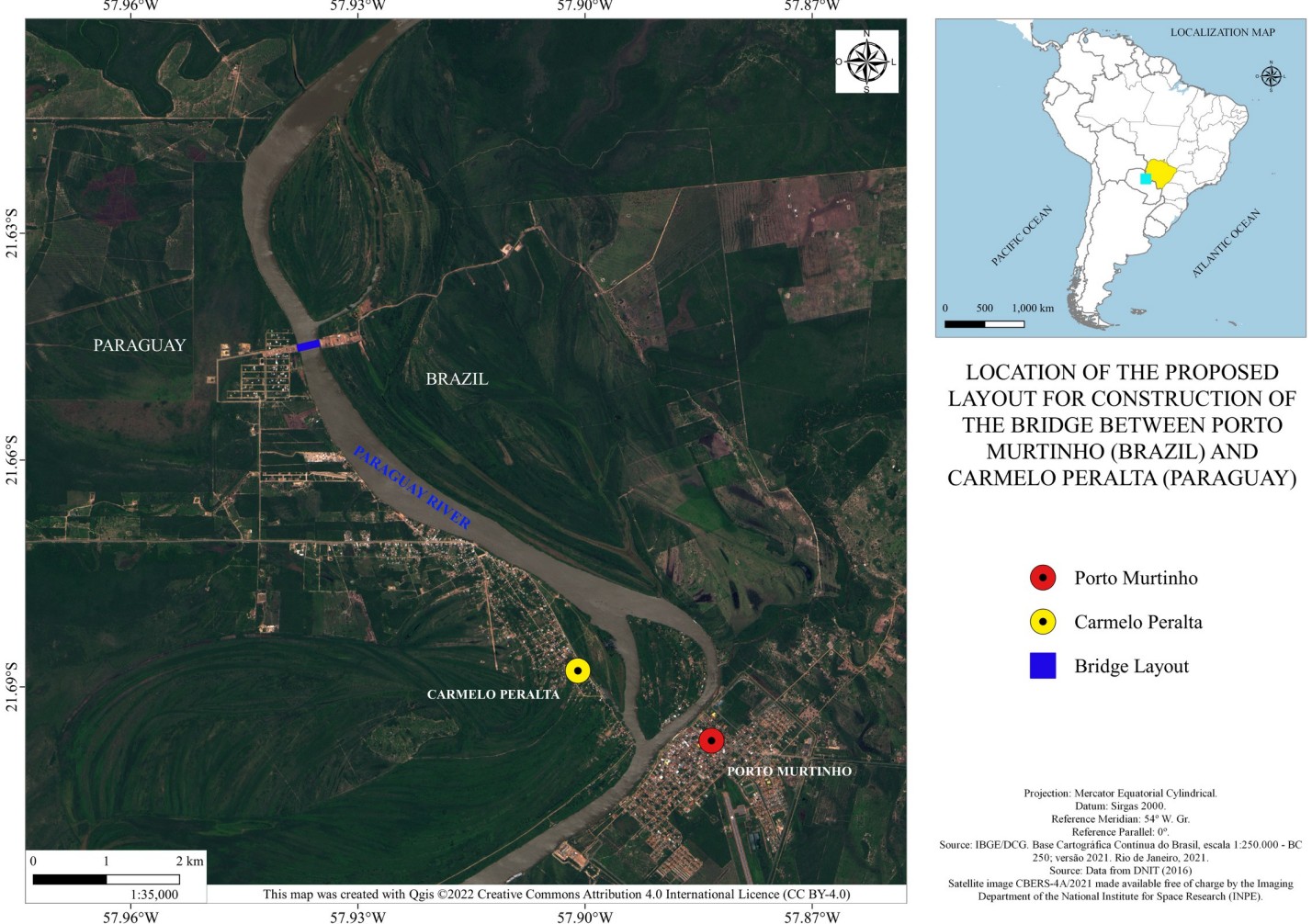

**Fig 2. Location of the proposed layout for the construction of the bridge linking Porto Murtinho (Brazil) and Carmelo Peralta (Paraguay).** Caption credit: Elaborated based on information from the [9, 13].

the Atlantic Ocean to the Pacific Ocean, we will most likely have the route with the entry into the State of Mato Grosso do Sul through the municipality of Três Lagoas, where there are the municipalities of Água Clara, Ribas do Rio Pardo, arriving in Campo Grande. Also, another possibility would be the entry through Bataguassu, passing through the municipalities of Nova Alvorada do Sul and Campo Grande—MS.

Given the above and following this sequence of municipalities where the origin is in the State of São Paulo, the Bioceanic Corridor has two main possibilities (Três Lagoas and Bataguassu), with a territorial unification of the route in Campo Grande, leaving Mato Grosso do Sul from the municipality of Porto Murtinho. Thus, the magnitude of the project is evidenced in the number of territories involved. The discussions on the regional economy and its role in territorial development will follow.

By analyzing the Local Productive Arrangements (LPAs) of the State of Mato Grosso do Sul, according to the [17], we can observe the significant importance of the municipalities of Campo Grande and Dourados. Moreover, most arrangements are closely related to primary production. A prominent issue is the absence of productive arrangements in the municipality

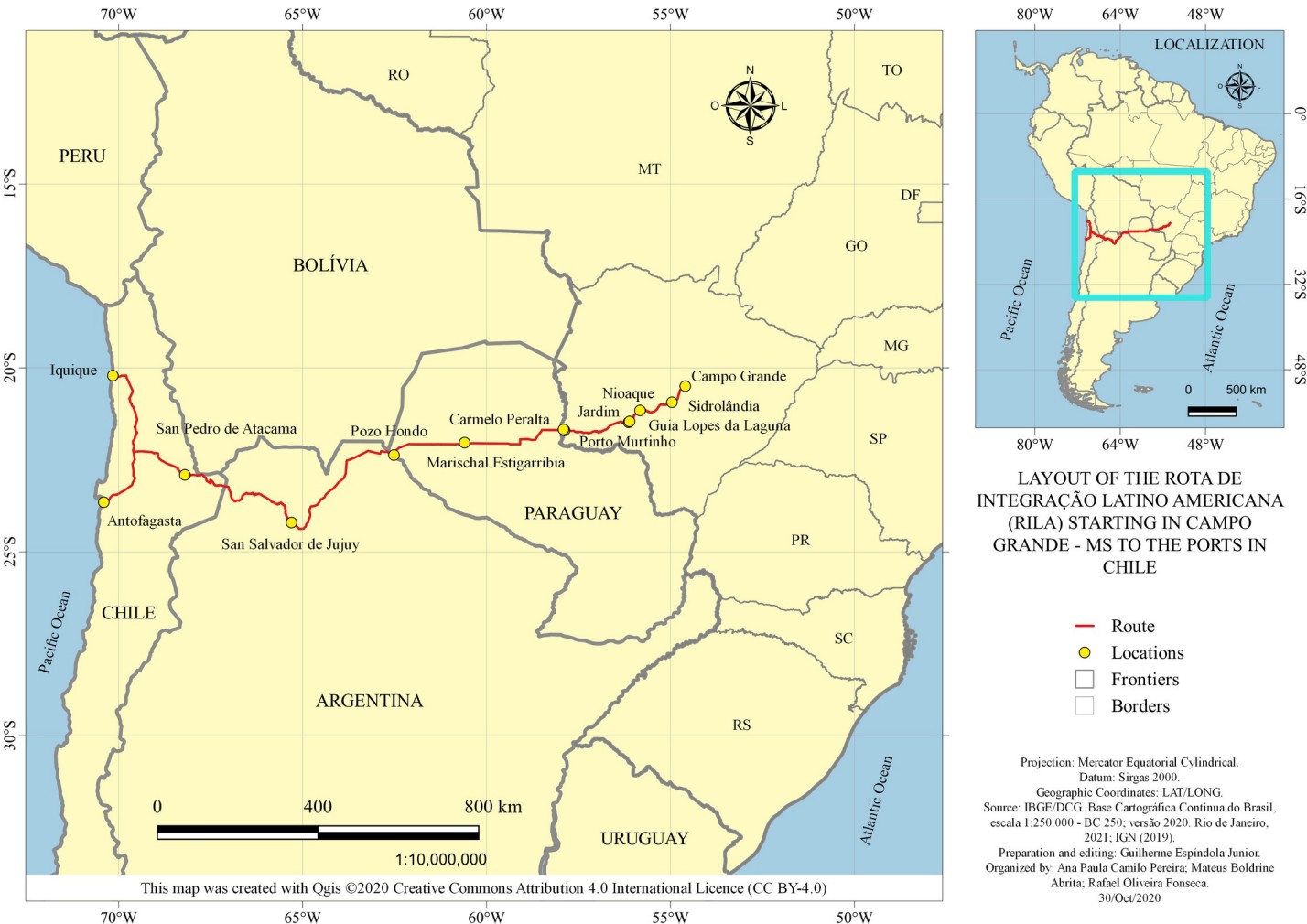

**Fig 3. The layout of the _Rota de Integração Latino Americana_ (RILA) starting in Campo Grande—MS to the ports in Chile.** Caption credit: Elaborated from [15].

of Porto Murtinho. This may indicate the importance of attention in relation to public policies to promote employment and income in that region.

At present, the most productive sectors in the state are livestock, fishing and aquaculture, textiles and clothing, wood and furniture, and tourism. This fact is added to the analyses presented earlier and reveals that the productive sector of the State has a strong role in the primary sector and natural resources. Fig 5 represents this characteristic, presenting the distribution of LPAs in the territory of the State of Mato Grosso do Sul.

Local productive arrangements that are strictly on the route can be seen on the map above. There are important LPAs that are in the vicinity of the route, such as beekeeping, agro-industrial horticulture, medicinal and herbal plants, manioc-culture, forest base, milk, and fish farming, all linked to primary production. A better understanding of the characteristics of spatial concentration can further contribute to the advancement of these LPAs from the implementation of the route.

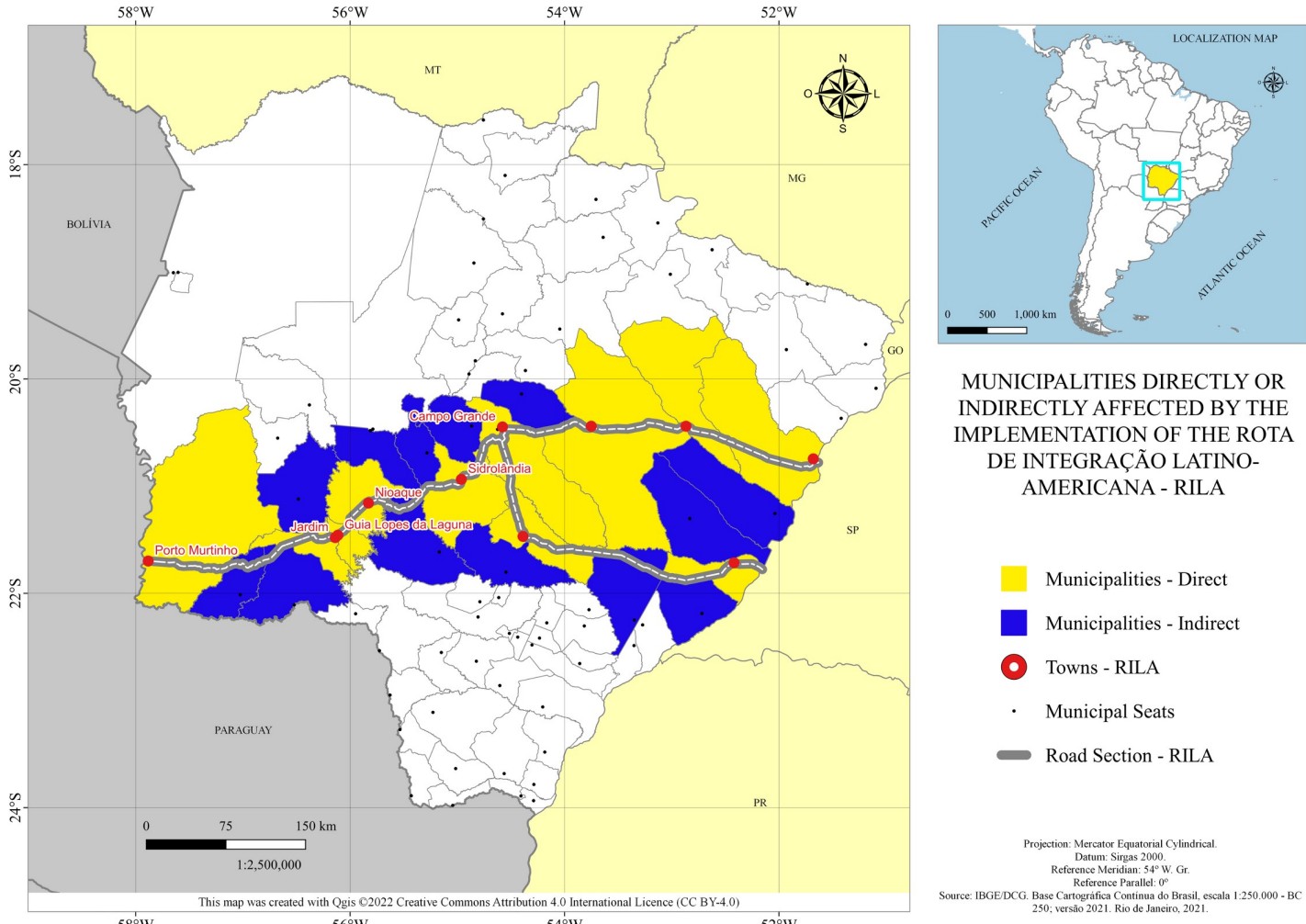

**Fig 4. Municipalities directly or indirectly affected by the implementation of the *Rota de Integração Latino Americana* (RILA).** Caption credit: Own development based on data from [9].

## Methodology and database

To have a better understanding of the relations with economic interfaces that the new international corridor can generate, it is important to perform a fine analysis of the productive concentration. To do this, first, it is necessary to define the criteria for the study of the productive concentration. Thus, the Composite Concentration indicator, proposed by [18], will be modified. It can capture four characteristics of an LPA, which is to say 1) the specificity of a sector within a region; 2) its weight in relation to the sector structure of the region; 3) the national importance of the sector; and 4) the absolute scale of local productive activities.

This index will be based on the existing knowledge of spatial techniques in conjunction with the Methodology of Composite Indicators [19], resulting in a spatially weighted Composite Concentration Index (*ICCs*). The importance of building ICCs can be revealed through two propositions: (a) one of the most evident characteristics of an economic scenario, as a whole, is the strong territorial concentration of economic activities, which is present in most countries and on various geographical scales, and (b) the fine analysis of spatial concentration is a

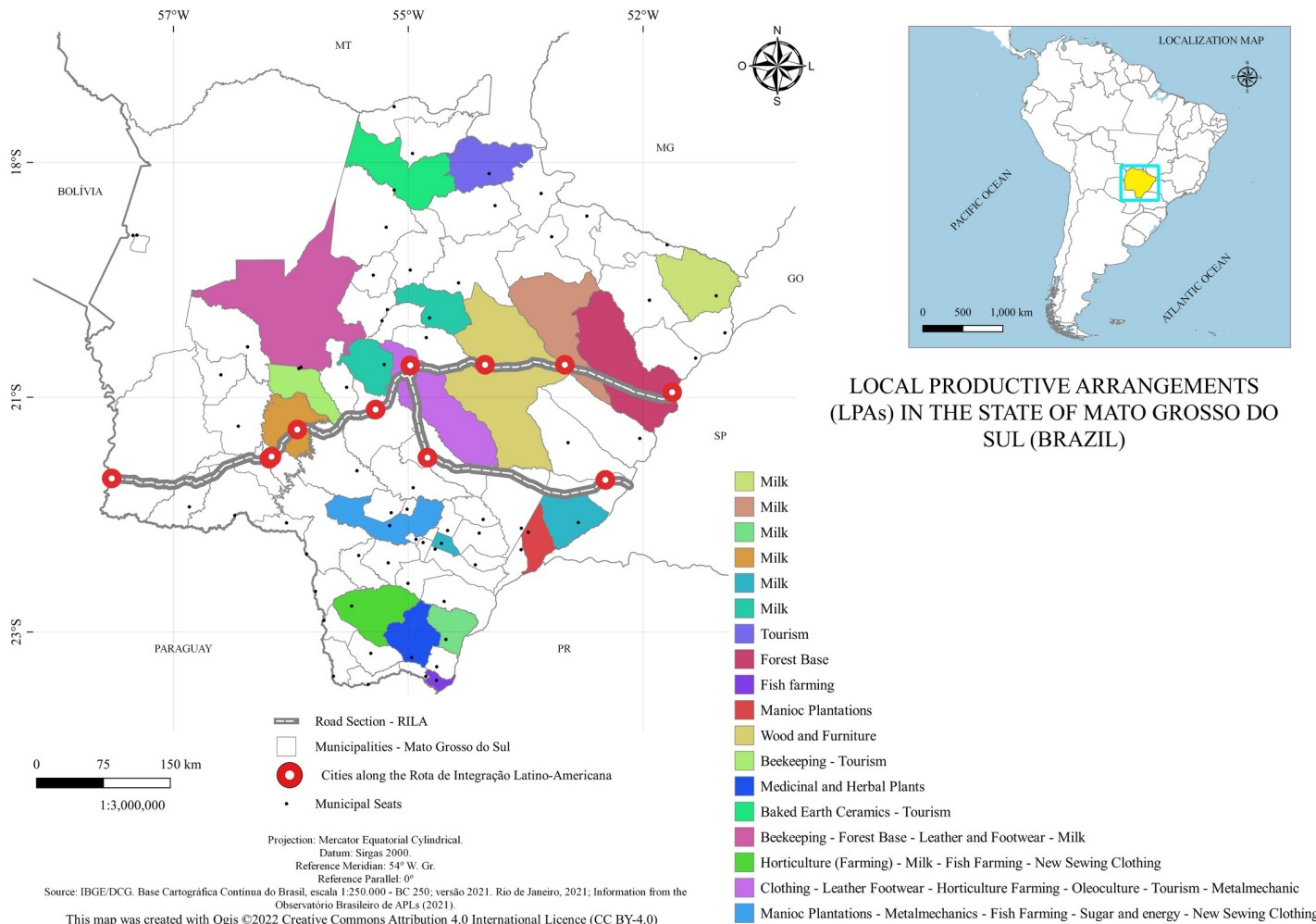

**Fig 5. Local Productive Arrangements (LPAs) in the State of Mato Grosso do Sul (Brazil).** Caption credit: Elaborated based on information from the [17].

prerequisite for other analyses that are concerned with understanding the dynamics of the various productive sectors of an economy, locally as well as nationally.

The modification that will be made in this indicator thus includes inserting the spatial dimension in the measurement, that is, formally adapting the index, so that it is spatially weighted (including algebraic articulations through a weighted average of neighboring values or spatial lags). Understanding the organization of activities, their disposition in the territory, and the recognition that they may have economic significance makes it possible to classify different concepts of region and territory, as well as the differentiation in relation to the concept of space. For further clarification on the importance of considering space in the analysis, see the theoretical discussions of [20–24], among others.

The Locational Quotient ($QL$), traditional in the literature of regional economics, compares two sectoral structures. The Quotient is the ratio between two economic structures: in the numerator, we have the "economy" under study, and in the denominator, a "reference economy." Its origin as an indicator of location and productive specialization occurred in the work of [25], which has been widely used in regional economic studies. It is an indicator applied to determine the degree of specialization in the production of a region or municipality in a

specific activity and can be represented as:

$$QL_{i,j} = \left( \frac{\frac{E^i_j}{E_j}}{\frac{E^i_{BR}}{E_{BR}}} \right) \qquad (01)$$

Where $E^i_j$ is the employment of the sector $i$ in the Region $j$; $E_j$ is the total employment in the region $j$ $E^i_{BR}$; is the employment in the Sector $i$ in Brazil; and $E_{BR}$ is the total sector employment in Brazil. When $QL_{i,j} = 1$, the specialization of the region $j$ in activity $i$ is identical to the specialization of the reference region (in this case, the aggregate of the regions) in that activity $QL_{i,j} = 1$; is the specialization of region $j$ when activity $i$ is lower than the specialization of all regions in this activity; and $QL_{i,j} > 1$ is the specialization of region $j$ when activity $i$ is higher than the specialization of all regions in that activity. The spatially weighted version of Eq (1) can be expressed by:

$$QLs_{i,j} = \left( \sqrt{QL_i} \right)' \Psi \left( \sqrt{QL_i} \right) \qquad (02)$$

Where $QL_{i,j}$ is defined in Eq 1 and is an array of spatial weights with generic elements $\psi_{i,j}$, and not null on the main diagonal. The array $\psi$ is designed to consider the repercussions that extend outside the boundary of the area considered. It can be built in many different ways, but for this case, it will be considered $\psi = I + W$, in which $W$ is a conventional weight matrix, standardized in the lines and with null main diagonal, and $I$ is an identity matrix of the same order.

The choice of which array of spatial weights to consider is an important decision in spatial studies because it is possible to generate spatial lags and the average of borderline values from them, determining which values will have, on average, their neighbors. For the application of spatially weighted indexes, *QLs*, *HHms*, and *PRs*, the matrix of contiguous binary space weights of the queen type, was used. This choice was guided by the fact that this type of array defines the neighbors of a locality considering its connection through the shared border or vertex. As the object of the analysis was done on the municipal scale, it is believed that this matrix format is the most appropriate, as it does not present a very expressive distortion in the values that make up such a matrix of spatial weights, the way a distance matrix would present, for example.

It is therefore valid to rewrite the spatially weighted productive specialization index (*Qls*) as follows:

$$QLs_{i,j} = QL_{i,j} + \left( \sqrt{QL_i} \right)' W \left( \sqrt{QL_i} \right) \qquad (2.1)$$

Note that the term, $\left( \sqrt{QL_i} \right)' W \left( \sqrt{QL_i} \right)$ is the productive specialization of this economy considering only the effect of neighborhood (existence of neighbors) in the analysis. The *QLs* index is the Conventional Location Quotient (*QL*) added to the weighted average of spatial interaction in the calculation. It is important to emphasize that if the neighbor relationship between the regions is not considered, so that $\psi = I$, or in the case of regions without neighbors (e.g., islands), the *QLs* value will be exactly that of traditional QL.

The second indicator that seeks to capture the real significance of the sector's weight in the local productive structure is the modified Hirschman-Herfindahl index (*HHm*), shown in [18]

as:

$$HHm_{i,j} = \frac{E_j^i}{E_{BR}^i} - \frac{E_j}{E_{BR}} \tag{03}$$

Where $E_j^i$ is the employment of sector $i$ in region $j$; $E_{BR}^i$ is the employment of the sector in Brazil; $E_j$ is the total employment in the region $j$; and $E_{BR}$ is the total sector employment in Brazil.

This indicator allows comparing the weight of activity $i$, region $j$, and activity $i$ of all regions in relation to the weight of the productive structure of region $j$ in the structure of all regions. A positive number indicates that the activity $i$ of region $j$ is more concentrated in region $j$ and, therefore, with greater power of economic attraction, given its specialization in such activity, it is more than in all other regions.

To avoid measuring indicators with negative numbers and allow the calculation of the square root of the indicator, the normalized version of the modified Hirschman-Herfindahl competitive concentration index ($HHm$) will be used:

$$HHm_n = \frac{(HHm + 1)}{2} \tag{3.1}$$

Its spatially modified version ($HHms$) can be written as:

$$HHms_{i,j} = \left(\sqrt{HHm_n}\right)' \psi \left(\sqrt{HHm_n}\right) \tag{4}$$

It is thus valid to rewrite the spatially weighted productive competitive concentration index ($HHms$) as follows:

$$HHms_{i,j} = HHm_{.j} + \left(\sqrt{HHm_{.j}}\right)' W \left(\sqrt{HHm_{.j}}\right) \tag{4.1}$$

Note that the term, $\left(\sqrt{HHm_{.j}}\right)' W \left(\sqrt{HHm_{.j}}\right)$, is the productive competitive concentration of this economy considering only the neighborhood effect (the presence of neighbors) in the analysis. The $HHms$ index is the Standardized traditional modified Hirschman-Herfindahl ($HHm_n$) plus the weighted average of spatial interaction in the calculation. As well as the $QLs$, where the neighborhood relationship between the regions is not considered, so that $\psi = I$, or in the case of regions without neighbors (e.g., islands), the value of the $HHms$ will be the same as $HHm$ presented by [18].

The third indicator traditionally known in the regional literature is the Relative Participation Index ($PR$) is able to capture the importance of the activity $i$ of the municipality $j$ in view of the total employment in the said activity for the other regions. The Eq is presented by [18] as:

$$PR_{i,j} = \left(\frac{E_j^i}{E_{BR}^i}\right) \tag{5}$$

Where $E_j^i$ is the employment of sector $i$ in region $j$ and $E_{BR}^i$ is the employment of sector $i$ in Brazil. This indicator varies between zero and one and, the closer to one, the greater will be the importance of the activity $i$ of the municipality $j$ in the other regions. The spatially modified

Relative Participation indicator (*PRs*) may be shown as:

$$PRs_{i,j} = \left( \sqrt{(PRs_i)} \right)' \psi \left( \sqrt{(PRs_i)} \right) \qquad (6)$$

It is valid to rewrite the PRs in the following way:

$$PRs_{i,j} = PR_{i,j} + \left( \sqrt{(PR_i)} \right)' W \left( \sqrt{(PR_i)} \right) \qquad (6.1)$$

Note that the term, $\left( \sqrt{(PR_i)} \right)' W \left( \sqrt{(PR_i)} \right)$, is the relative participation of this economy considering only the neighborhood effect (the presence of neighbors) in the analysis. The PRs index is the conventional relative participation PR index plus the weighted average of the spatial interaction in the calculation.

These three indices provide the necessary parameters for the development of a composite concentration indicator called *ICCs*. For its calculation—for each sector of activity and geographical unit under study—we propose to perform a linear combination of the three standardized indicators (Eq 7). Therefore, each of the three indexes, used as *ICCs*, may have a distinct capacity to represent agglomeration forces, especially when it is possible to consider the various sectors of the economy.

For the next step, it is necessary to calculate the specific weights for each of the supplies in the productive sectors:

$$ICCs_{i,j} = \alpha QLs_{i,j} + \beta HHms_{i,j} + \theta PRs_{i,j} \qquad (7)$$

Where $\alpha$, $\beta$ and $\theta$ are the weights of each of the spatially weighted indices for each specific productive sector of the Brazilian economy.

The standardization of the three indexes that make up the Spatially modified Composite Concentration Index (*ICCs*), namely *QLs*, *HHms*, and *PRs*, it is necessary and consists of subtracting from a value of a variable (or values of the previously calculated indexes) its average and dividing the result by the standard deviation of the set or variable. Thus, the standardization corresponds to shifting the center (given by the average) from a data set to the origin of the Cartesian system.

The decision to choose the best method will be made after considering the results obtained with the application of some more common methodologies, which are Factor Analysis and equal value weighting, i.e., for all indicators will be given the same weight (a simple average of its three "subindices," one-third each). Of the techniques measured, the one with the most satisfactory results and better adjustment was the rotation of the main components. In summary, positive numbers for ICCs mean that for a sector of a specific municipality, the concentration of economic activity is higher than that observed in the average for the State of Mato Grosso do Sul. Negative values, on the other hand, denote a concentration lower than the state average for the activity under analysis for the municipality.

The data used to estimate the *ICCs* were obtained from primary sources of the Federal Government of Brazil. In particular, data from the Program for the Dissemination of Labor Statistics (PDET) of the Ministry of Labor was used, referring to the Annual List of Social Information (RAIS) [26], as well as the classification of these data according to the large sectors of economic activity of the Brazilian Institute of Geography and Statistics (IBGE), namely Industry, Civil Construction, Trade, Services, and Agriculture.

Data was collected for the year 2020, with the most recent periodicity available in the database. The information was collected for the number of formal jobs and the numbers of employers, according to the five major sectors previously pointed out. The geographic scale

defined was the municipal scale, with the information collected being analyzed for the 79 municipalities of the State of Mato Grosso do Sul.

## Analysis of weighted composite concentration index

From this fine analysis of the concentration, it will be possible to explore the productive specialization of the State, the degree of concentration of economic activities in the municipalities, and the level of relative importance among the selected sectors, namely Industry, Construction, Trade, Services, and Farming. It is important to note that the methodologies used differ from those of the [17]. The difference is in the metrics of identification of the arrangements since, as it is for the most of the work in Regional Economics, the federal institution uses the Locational Quotient in its traditional form.

This work goes beyond the classical parameters in the sense of inserting the spatial issue and its unfolding, spillovers, and neighborhood effects, with the application of the matrix of queen weights. This matrix format has a space provision that allows a municipality to have more than one LPA, so the joint information between concentration and productive specialization is important and will be considered in the following discussion.

## Results for spatially weighted composite concentration index

To better understand the concentration of productive activities in the State of Mato Grosso do Sul, the spatially weighted Composite Concentration Index (*ICCs*) was calculated, considering its formation of the levels of productive specialization (*QLs*), competitive concentration (*HHms*), and relative participation (*PRs*). The *ICCs* was calculated considering the municipal spatial scale listed for the following sectors: Industry *ICCs1*, Construction *ICCs2*, Trade *ICCs3*, Services *ICC4*, and Farming *ICCs5* for the year 2020.

Initially, the analysis is engendered by presenting the reliability test of the data, which is formed by the subindexes of spatially weighted regional analysis of productive specialization (*QLs*), modified Hirschman-Herfindahl competitive concentration (*HHms*), and relative participation (*PRs*). The adequacy test of the applied data was the KMO (*Kaiser-Meyer-Olkin*). This is presented in Table 1 in the Schedule and is a complementary part of this analysis. Through it, it was possible to notice that the data was consistent and significant for the analysis that is later constructed. In most of the sectors considered—that is, Industry, Civil Construction, Trade, Services, and Farming—it was equal to or greater than 0.50 and significant at 1% ($p < 1\%$), both for the variable employment and a number of establishments.

Thus, the correlation matrix is not an identity matrix, and the variables are not correlated. As a result of the application of the test, it is possible to affirm that the set of data used in the analysis is adequate and dependable for the application of factor analysis, and consequently, the measurement of the Concentration Index (*ICCs*).

After presenting the adequacy and reliability test of the territorial subindices selected for analysis, it is necessary to determine how many factors will be extracted and their respective percentages of variance explained. There are criteria for defining the number of main factors to consider. For this study, we chose to include only the most explanatory factor in the analysis, with Eigenvalue greater than 1 (Kaiser criterion). From Table 2, it is possible to identify factor 1 (F1) as the most representative and with numbers greater than 1; this characteristic indicates that the characteristic root is larger than the unit, the so-called Eigenvalues. The total variance explained considering factor 1 is high, above 50% for most productive sectors observed.

Once the factor that will be extracted for the analysis has been identified, it was necessary to evaluate the loadings factors, which are presented in Table 3. The standardization of the three

**Table 1. Kaiser-Meyer-Olkin measuring of sampling adequacy (KMO).**

| KMO | Employment | Establishments |
|---|---|---|
| **Sectors** | | |
| Industry | 0,30 | 0,47 |
| Construction | 0,50 | 0,49 |
| Trade | 0,54 | 0,50 |
| Services | 0,44 | 0,51 |
| Farming | 0,53 | 0,45 |

**Source**: Elaborated from secondary data extracted from RAIS/MTE (2021).

indexes that make up the Spatially modified Composite Concentration Index (*ICCs*)—*Qls*, *HHms*, and *PRs*—was performed based on varimax orthogonal rotation. Of all the methods assessed, the choice was made after analyzing the results obtained and after finding the most satisfactory loadings, adjusting it to the reality that is desired to be measured, and preserving the property of maximizing the variance of factor loadings.

In addition to the factor loadings, the commonalities are presented, that is, the demonstration of the joint explanatory capacity of the two factors in relation to each sub-indicator. As stated earlier, the portion explained by the common factors is called commonality, and the unexplained portion is called specificity.

Commonalities can range from 0 to 1; values close to 0 indicate that common factors do not explain variance, and values close to 1 indicate that all variances are explained by common factors. For these results, the commonalities indicate that all factors have their variability, significantly captured by the factor. According to the values of the commonalities, a more pronounced explanatory capacity was found for the subindex of the Locational Quotient (*QLs*), followed by the modified Hirschman-Herfindahl (*HHms*) and the Relative Participation (*PRs*). The provisions of factor loadings characterize the main factors of the analysis. In the spatial scale referring to the municipalities of Mato Grosso do Sul, the first factor (F1) is characterized by higher weights to the sub-index of the Locational Quotient (*QLs*), followed sequentially by the modified Hirschman-Herfindahl (*HHms*) and the Relative Participation (*PRs*). These sub-indicators reflect, respectively, the influence of the level of productive specialization acting mainly on the aspects of productivity.

**Table 2. Eigenvalues and the percentages of total variance explained for Factor 1 (F1) identified by the extraction of the main components.**

| 2020 | | Employment | Establishments |
|---|---|---|---|
| **Industry** | Eigenvalue | 1.522 | 1.954 |
| | % Variance explained | 50.74 | 65.13 |
| **Construction** | Eigenvalue | 1.981 | 1.976 |
| | % Variance explained | 66.22 | 65.89 |
| **Trade** | Eigenvalue | 1.542 | 1.951 |
| | % Variance explained | 51.36 | 65.01 |
| **Services** | Eigenvalue | 1.964 | 1.978 |
| | % Variance explained | 65.49 | 65.96 |
| **Farming** | Eigenvalue | 1.418 | 1.498 |
| | % Variance explained | 47.09 | 49.93 |

**Caption credit:** Elaborated from secondary data extracted from [26].

**Table 3. Factorial loads after orthogonal rotation and commonalities, obtained in factor analysis, which make up the *ICCs*, considering the municipalities of Mato Grosso do Sul in the different productive sectors.**

| 2020 | | Employment | | Establishments | |
|------|------|------|------|------|------|
| | | F1 | Commonality | F1 | Commonality |
| *Industry* | *QLs* | 0.751 | 0.936 | 0.697 | 0.512 |
| | *HHms* | 0.657 | 0.921 | 0.946 | 0.896 |
| | *PRs* | 0.383 | 0.903 | 0.803 | 0.652 |
| *Construction* | *QLs* | 0.826 | 0.973 | 0.975 | 0.832 |
| | *HHms* | 0.499 | 0.971 | 0.988 | 0.501 |
| | *PRs* | 0.501 | 0.979 | 0.994 | 0.511 |
| *Trade* | *QLs* | 0.738 | 0.545 | 0.821 | 0.577 |
| | *HHms* | 0.548 | 0.676 | 0.967 | 0.936 |
| | *PRs* | 0.528 | 0.723 | 0.944 | 0.893 |
| *Services* | *QLs* | 0.894 | 0.848 | 0.946 | 0.883 |
| | *HHms* | 0.507 | 0.934 | 0.984 | 0.518 |
| | *PRs* | 0.503 | 0.955 | 0.988 | 0.501 |
| *Farming* | *QLs* | 0.652 | 0.876 | 0.865 | 0.755 |
| | *HHms* | 0.588 | 0.785 | 0.846 | 0.717 |
| | *PRs* | 0.553 | 0.597 | 0.804 | 0.646 |

**Caption credit:** Elaborated from secondary data extracted from [26].

Theoretically, the spatially weighted Locational Quotient (*QLs*) can act positively on productivity through economies of scale and agglomeration originated by greater specialization and locational concentration, in addition to capturing the average interactions between neighbors, referred to by the international literature as neighboring effects [27].

In addition to this first factor, spatially weighted Hirschman-Herfindahl sub-indices (*HHms*) are capable of capturing the real weight of an economic activity in the productive structure of a locality. According to Crocco et al. (2003), it is worth noting that in the literature on regional economics, the sub-indicator (*QL*) is more appropriate for measuring impacts in medium-sized regions. For small regions, with low industrial employment and poorly diversified productive structure, the Locational Quotient tends to overvalue the weight of a given sector for the region. Similarly, the quotient also tends to undervalue the importance of certain sectors in regions with a more diversified productive structure, even if this sector has significant weight in the context of the state. To mitigate this problem, the modified Hirschman-Herfindahl sub-indicator (*HHms*) was used, which was designed to capture the real significance of the sector's weight in the local production structure, followed by the Relative (*PRs*).

For an analysis at the municipal level, as well as highlighted by [28], it is expected that the productive specialization and competitive concentration will further evince gains from a theoretical definition with external economies and the levels of competitiveness, such as the advantages originated through the locational concentration, the improvement of processes and products, with availability and quality of raw materials and supplies close to sources, in addition to the operational practices of business administration and joint actions that influence the level of competitiveness, both in the domestic market and in the exporting sectors and that compete internationally, as is the case of Industry and Farming, for instance.

In addition, another important factor that influences external economies is the technical progress articulated by technological innovations, both in products and in processes, whose rapid dissemination to a set of conglomerate companies and the development of activities in the same sector (as they are close to other companies and specialized professionals) creates

favorable conditions for the increase in their economic activities and the development of new businesses.

In an intersectoral temporal analysis, the possible changes occurred because of the way economic activities are articulated with each other and with the medium itself via backward and forward linkages. The productive chains (chain effects) that these activities can generate stimulate the boosting of resources, capital, investments, and the process of economic growth. According to [29, 30], this chaining process and its ability to generate more economic development lead to political and economic strategies that help "reflect" on the main obstacles arising from underdevelopment and inequalities in several countries and regions. The backward threads come from an autonomous growth of a given sector (in our study, Industry, Farming, and Civil Construction), stimulated basically due to new investments or the use of pre-existing productive capacity.

This chaining induces the growth of other sectors related to it through "demand pressures." Forward threads, on the other hand, are derived from an increase in the production of a certain factor that favors the increase in the production of other sectors due to an oversupply of the initial sectoral product [29, 30].

These threads can also be captured from the loadings factors matrices of the following sectors studied: Industry, Farming, and Construction, in which, the sectors with the higher/lower intensity in their bonds compared to others have an increase/decrease in the value of their spatially weighted sub-indicators ($Qls$, $HHms$, and $PRs$), according to the matrix composition of their loadings factors.

After this detail that evinces the robustness of the methodology applied and the possible paths of analysis provided, the spatialized data of the ICCs for each of the economic sectors will be presented. ICCs are presented in a comparative manner: blue for employment data and green for data on the number of establishments.

The first analysis that extends to all sectors analyzed is that certain differences in the spatial concentration of each sector are remarkable for different data. This difference has an important result, in which certain municipalities have a concentration of companies, but this concentration does not occur with jobs, but that the opposite is true. This result, although obvious by the difference in databases, demonstrates that the concentration of economic activity has distinctive characteristics, such as the size of firms, the number of jobs generated by each firm, and the intensity of use of technology and labor, all of which are examples that contribute to explain these differences.

Mato Grosso do Sul has a base of establishments with 87% individual microentrepreneurs and microenterprises considering branches and headquarters, according to data from the Internal Revenue Service for the year 2020. Individual microentrepreneurs can employ only one person, while micro enterprises are classified with up to nine employees in the trade and services sector and up to 19 employees in the industrial sector.

If added to the small enterprises, which are thus classified as 10 to 49 persons in trade and services or 20 to 99 persons in the industry, we have a total of 93% of the companies in the State. This will therefore be the representative agent of the analysis, clearly justifying the differences in concentrations between the analysis of establishments and jobs, and knowing that the average company will generate, on average, a number of jobs greater than one, we will put more emphasis on the analysis of concentration by employment data, which is the commonly used basis in literature.

An additional point to this difference in concentration between the employment databases and establishments refers to the pattern observed in the State of Mato Grosso do Sul, in which small municipalities have a smaller number of establishments compared to Campo Grande, Dourados, Três Lagoas, and Ponta Porã.

The darker colors of the images represent municipalities with a higher concentration of companies with jobs in each of the sectors analyzed; in these cases, the number of the indicator is positive, which indicates concentration above the state average. Intermediate colors show numbers close to zero and concentrations that are less intense or very close to the state average, while lighter colors represent municipalities with almost no concentration, which is below the state average.

For the industrial sector, observed in Fig 6, it is possible to note that the eastern region and the entire eastern border territorial strip of the State in this region present a concentration of industrial sector jobs above the state average. The municipalities with the most intense concentration are Mundo Novo, Eldorado, Itaquiraí, Iguatemi, Juti, and Vicentina in the extreme south of the state. To the southeast, the second set of municipalities with a higher-than-average concentration are Batayporã, Anaurilândia and Bataguassu, Nova Andradina, Angélica, and Rio Brilhante.

In the northeast of the State, we have a third set of municipalities composed of Paranaíba, Aparecida do Taboado, and Selvíria. Among these municipalities with the highest concentration of the industrial sectors, the activities that stand out are dairy products, clothing accessories and underwear, food production usually associated with the processing of meat products, production of oils, or the processing of products linked to farm production. The pattern of industrial sectors that have concentration above the state average in some municipalities are usually associated with agriculture, and are concentrated in general, in municipalities producing supplies and have the availability of labor, which is of average pay or below the state average, which is currently R$ 2,757 reais (BRL). This amount correspond to the usual average income of the main labor of persons 14 years of age or older, employed in the reference week in formal work [31].

This type of industry associated with the agricultural sector has a rule of allocation of production plants, usually based on the distance from the raw material, and in the case of the municipalities observed that present concentration issue, this is true. The case of dairy products, for example, given the perishability and difficulty of transport over long rural roads due to the need to refrigerate milk, it makes no sense for the industry to be far from the raw materials.

Another point is that this type of industry needs to ensure a minimum quantity of supply volume to operate production, and this is also a deciding factor for plant allocation. In this sense, it is natural that municipalities, such as Mundo Novo, Itaquiraí, Batayporã, Anaurilândia, and Rio Brilhante—which show above-average concentration in the industry, with the activity of dairy products as one of the outstanding industrial activities—have a livestock production based on the production of milk.

Fig 5 presenting the LPAs of Mato Grosso do Sul, according to the [17], shows some of the municipalities verified here with a high concentration of the industrial sector, especially dairy products, as in the case with Anaurilândia, Itaquiraí, and Bandeirantes, which are not among the municipalities with the highest concentration in the industrial sector, but have a concentration higher than the state average and several dairy production plants.

The activity of the production of clothing, accessories, and underwear also appears as some of the most representative activities of the industrial sector in the most concentrated municipalities of the industrial sector. These plants typically use the criterion of allocation of their plants based on the availability of paid labor with low yields, since they are the products of low-added value in general and compete via prices without the possibility of determining them for large product differentiations. For this reason, these companies need easily replaceable and inexpensive labor.

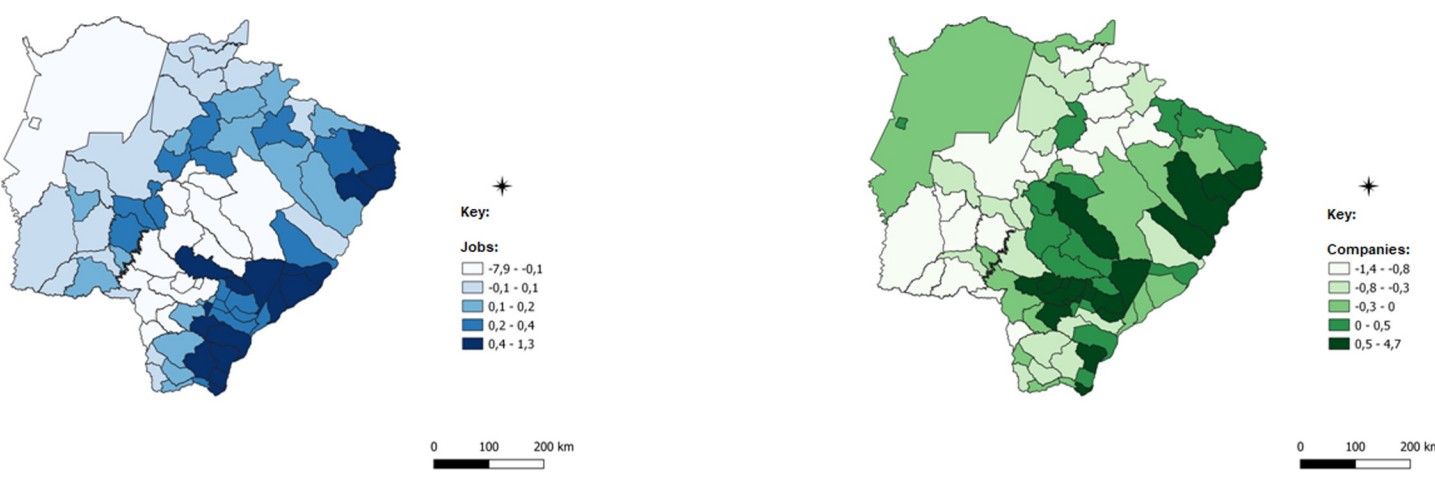

**Fig 6. Industry spatially weighted composite concentration index.** Caption credit: Elaborated from secondary data extracted from [26].

One point that draws attention is that, in general, the manufacturing industry usually has a very flexible plant, with low capital content, which allows relocations, if necessary. The search for labor could be one of the factors that can generate this relocation, or even fiscal incentive policies and subsidies that guarantee competitiveness in terms of prices to these industries. Unlike dairy, there is nothing that fixes this type of industry in one place. Among the twelve municipalities with greater concentration of the industrial sector, eight of them are big on clothing activities, including the production of footwear and accessories, underwear, clothing in general, or in the production of textile articles. These municipalities include Mundo Novo, Eldorado, Iguatemi, Juti, Batayporã, Nova Andradina, and Rio Brilhante.

Among the other municipalities, Campo Grande, Três Lagoas, and Dourados have many establishments in industrial manufacturing activities; however, these agents must have characteristics of the representative company previously presented, which makes them on average small-sized and generate only a few jobs per manufacturing unit. Along with other activities of the industrial sector in the generation of jobs, these municipalities, however, were not apparent in ICCs calculated by employment. Other activities that were highlighted among the municipalities with the highest concentration in the industrial sector were ceramic artifacts in

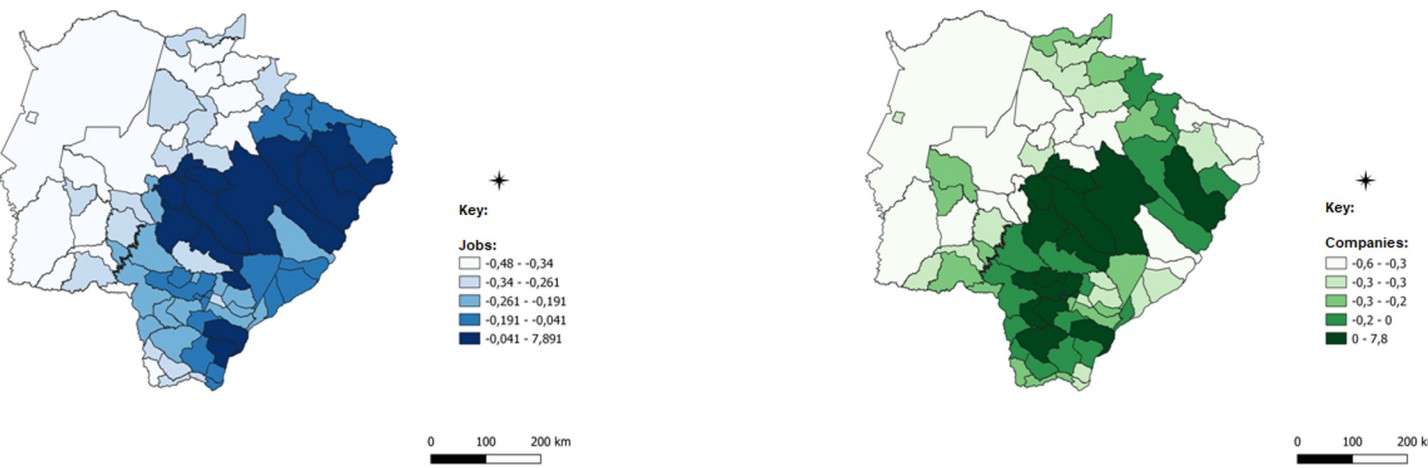

**Fig 7. Spatially weighted Compound Concentration Index of civil construction.** Caption credit: Elaborated from secondary data extracted from [26].

Eldorado, which is traditionally a region dedicated to this type of production, given the availability of natural resources. The manufacturing of furniture in Mundo Novo, metalworks in Itaquiraí, and metallurgy in Vicentina, Bataguassu, and Nova Andradina are other examples.

The construction sector, observed in Fig 7, presents the concentration in the central municipalities of the State, especially around Campo Grande, the capital of the State. There is a strip of land that extends to the east coast of the state. The municipalities there include Campo Grande, Sidrolândia, Nova Alvorada do Sul, Terenos, Rochedo, Jaraguari, Ribas do Rio Pardo, Água Clara, Brasilândia, Inocência, Três Lagoas, Selvíria, and Aparecida do Taboado. The region around the municipality of Três Lagoas is one of the regions producing ceramic products for civil construction (e.g., bricks, roof, floor, and wall tiles).

The municipalities with the highest concentration of the commercial sector within their territory, observed in Fig 8, considering the employment data were Campo Grande, Sidrolândia, Nova Alvorada do Sul, Rio Brilhante, Ribas do Rio Pardo, Jaraguari, Rochedo, and Terenos in the center of the State. In the eastern region of the State, further north, there is a second set of municipalities that presented high ICCs for this sector: Paranaíba, Aparecida do Taboado, Selvíria, and Três Lagoas. Three other municipalities on the eastern side of the state further south also showed high concentration of trade, which are Bataguassu, Itaquiraí, and Eldorado.

When considering the number of establishments, some municipalities lose importance, but a range of municipalities around the region of Dourados and Três Lagoas gained prominence in the concentration of trade analyzed by ICCs. Despite having a considerable number of establishments in this sector, the companies of Dourados and Três Lagoas, and their surroundings, did not generate as many jobs in relation to the state average that the establishments of Campo Grande and the surrounding areas did.

It is possible to notice that for trade, there is a larger set of municipalities that have a concentration higher than the state average and that they are usually located around the areas with the largest ICCs. Further, the trade activities that stand out in the generation of formal jobs in municipalities with a high concentration are in retail: trade of articles for clothing and accessories, retail trade of merchandise (hypermarkets, minimarkets and warehouses, and especially supermarkets surpass the retail trade of clothing in a number of jobs), retail of automobiles, trucks, and utility vehicles, retail of accessories and parts for vehicles, building materials, and retail in pharmaceuticals and supplies for gas stations. Some activities are a little less significant

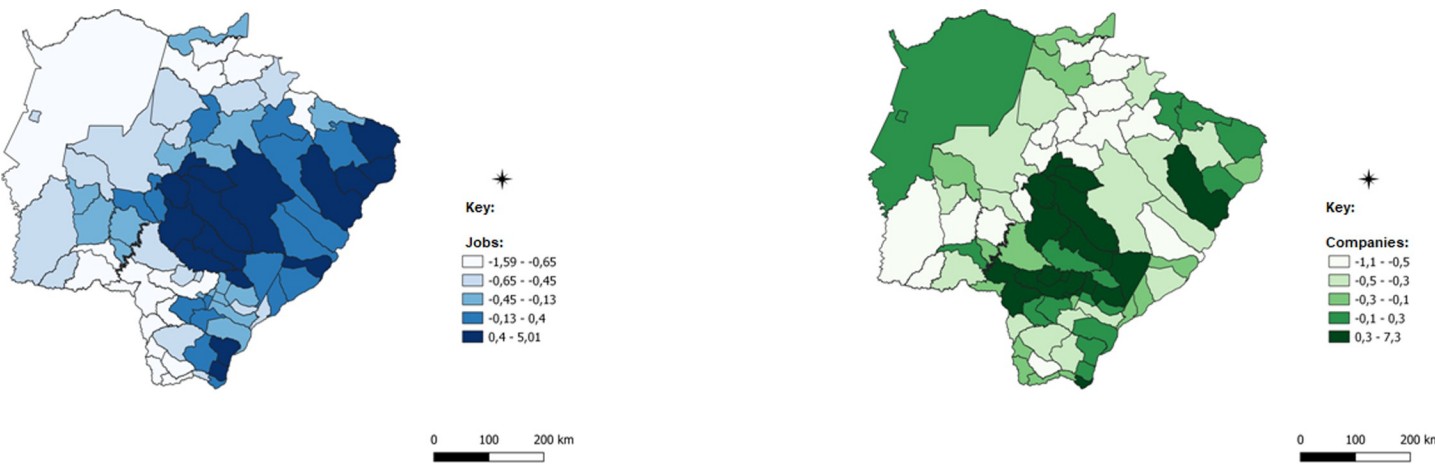

**Fig 8. Spatially weighted Composite Trade Concentration Index.** Caption credit: Elaborated from secondary data extracted from [26].

than those previously mentioned in the generation of jobs, but among those, the main ones are convenience, butcher, and tobacco stores.

Wholesale activities are those intricately linked to the supply trade, such as supermarkets and the distributors of parts and accessories for vehicles. There are variations among these activities in relation to the amounts paid, the salary, and the size of the companies. Wholesale activities and dealerships representing car retail, supermarkets, and hypermarkets tend to be larger companies with a larger number of employees, while the others fall within the characteristic of the representative agent.

The trade sector hires more individuals with full elementary and middle-education levels, and the average salary for these qualifications in the State is around R$ 1,584.48 reais (BRL) for individuals with full elementary education and R$ 1,804.20 (BRL) for individuals with a full high school education. In the case of trade, the activities in larger companies are mostly associated with the hiring of more qualified personnel.

Except for meat trade and some categories of food products, and for some cases of the trade in construction materials, the trade in Mato Grosso do Sul is not linked to local industries, that is, it has no connection in its supply chain with companies that produce within the State. In general, they are products purchased in other States, transported by road and, for most of the reported activities, they are linked to supply, that is, the consumers are local. This fact produces a large flow of entry of merchandise for the local supply.

In the services sector, shown in Fig 9, the municipalities with the highest ICCs considering the employment data were Campo Grande, Ribas do Rio Pardo, Jaraguari, Rochedo, Terenos, Sidrolândia, and Nova Alvorada do Sul. Further west, connecting some border municipalities near Paraguay, there is another set of municipalities with a high concentration of this sector, formed by Dourados, Douradina, Itaporã, Ponta Porã, Laguna Carapã, Fátima do Sul, and Deodápolis. The municipalities of Ladário and Bonito also show a high concentration of the services sector.

Bonito and Ladário have similar characteristics regarding the services sector. Bonito is one of the main ecotourism destinations in the country, tourism being one of the main economic activities of the municipality, and the services sector there is linked to lodging, food, and tourist intermediation. Its business base are hotels, inns, restaurants, bars, travel agencies, and tourist attractions. Ladário does not have the same tourist vocation but is considered an important destination for fishing and contemplation tourism, as it is in the Pantanal of Mato

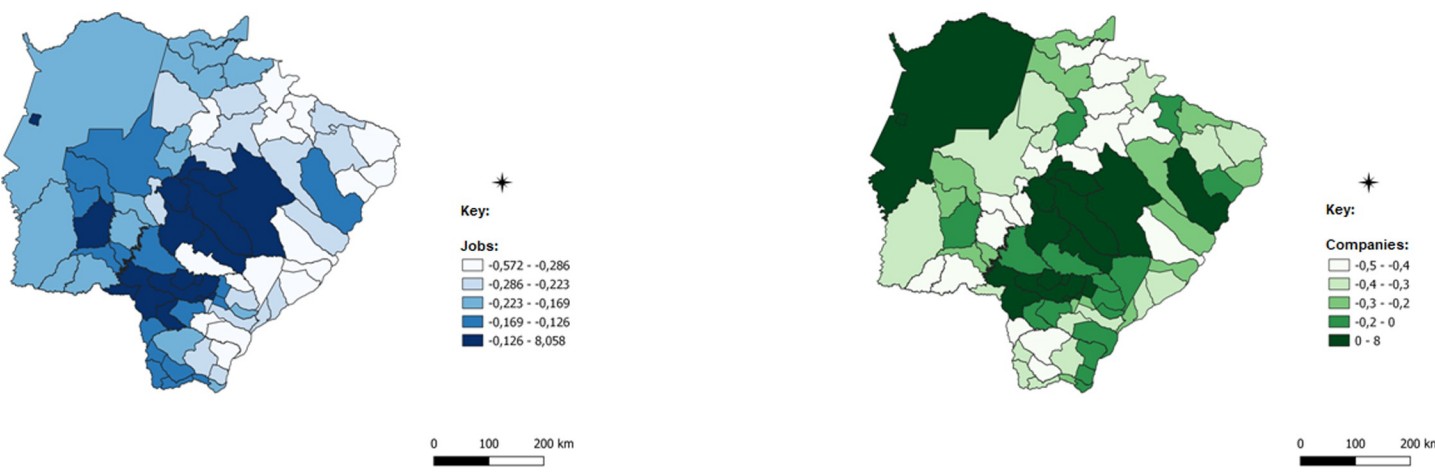

**Fig 9. Spatially weighted Composite Concentration Index of Services.** Caption credit: Elaborated from secondary data extracted from [26].

Grosso do Sul. In addition, it has the navy and military service linked to border control and this generates a continuous movement of military personnel, which contributes to the activity of hosting services. In addition, travel agencies and transport services on boats are also important activities in this municipality.

Of the other two sets of municipalities with concentration of the service sector, only Dourados and Campo Grande have a more complex service sector, with a more extensive set of activities that varies in the level of sophistication of the services provided and in the type of labor employed. Some standards verified in Dourados as main activities are private security, transportation of valuables, road transport of cargo, dental and hospital care services, restaurants, education at the fundamental, secondary, and higher levels, and storage services.

In general, these activities generate the highest numbers of jobs for the service sector of this municipality, and these are activities that require labor with additional technical training or even higher education. This is verified by analyzing the average pay for workers in the service sector of Dourados, with a full high school education of R$ 1,717.61 (BRL), with a full elementary education of R$ 1,469.94 (BRL), and with a full higher education of R$ 3,149.92 (BRL), according to [26].

While for the other municipalities, the average pay for individuals with elementary education was R$ 1,311.24 (BRL), for full high school education, it was R$ 1,546.90 (BRL) and for full higher education, it was R$ 2,367.95 (BRL); that is, especially for full higher education, the average salary is significantly lower, 33% [26].

The other municipalities have service sectors that are more dedicated to activities such as hotels, preschool education, storage, restaurants, and road and freight transport, which are closer to meeting the subsistence needs of the population, which when they need more complex and specialized services, go to Dourados to consume or hire workers, considering that the distance is very less between these municipalities.

In the case of Campo Grande, the main activities are road cargo transport, cleaning of buildings and households, telephone services, restaurants, snack bars and the like, private security, building support services, hospital care, education at the fundamental, middle, and higher levels, passenger transport, administrative and accounting services, banks and credit unions, lawyers, engineering services, hotels, outpatient and health care medical services, registration and collection activities, technical support services and technological maintenance, the selection and agency of labor, and the distribution of electricity.

The services sector adds more value to the gross domestic product of the municipality of Campo Grande than the industry itself. Due to the complexity of this sector, in Campo Grande, the population of the surrounding municipalities also use the services of this sphere, coming to Campo Grande. The municipality of Terenos or Jaraguari are about 30 or 40 minutes from the center of Campo Grande, which makes their movement common and Campo Grande a center in this sector.

Farming, as observed in Fig 10, is the sector with the highest number of municipalities, with concentrated or high ICCs or above the state concentration average. This fact is verified because, in general, the municipalities of the State of Mato Grosso do Sul have their productive structure historically based on farming, having as its main activities the ranching of beef cattle, pigs and poultry, dairy cattle, and egg production. In agriculture, soybean and corn crops were important in the economic formation of the State and persist as the main crops in volume, planted area, and gross production value.

In the mid-2000s, forest production intensified due to the implementation of industrial projects for the processing of wood and the production of bleached hardwood kraft pulp in Três Lagoas. Since then, forest production has grown in the state and has become one of the main crops of agriculture. Currently, Mato Grosso do Sul is among the six largest producers of

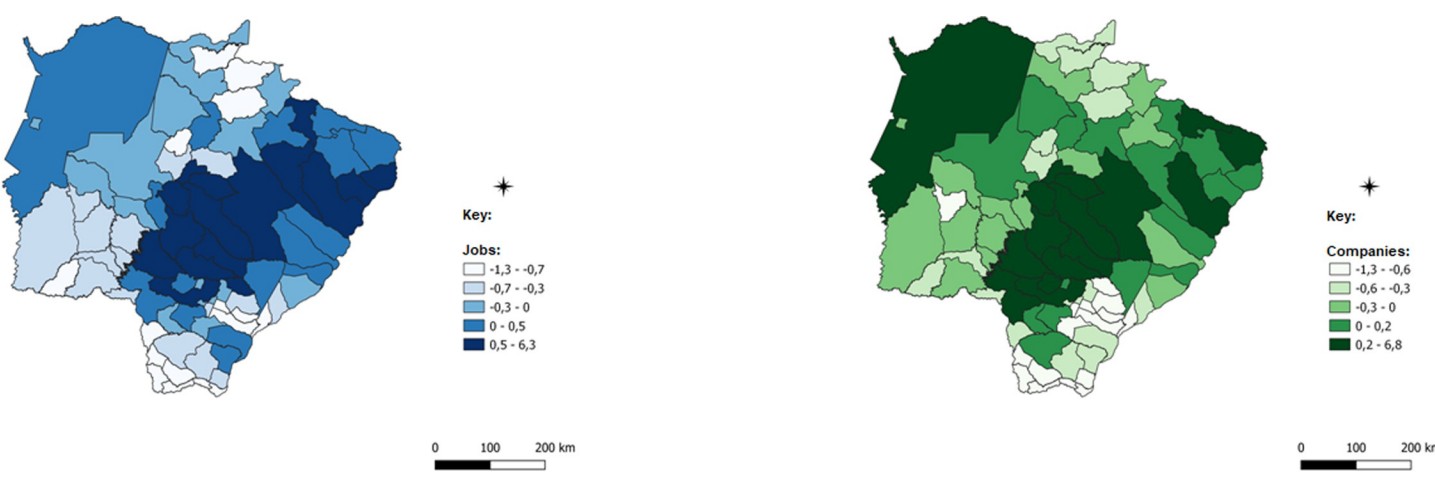

**Fig 10. Spatially weighted Compound Concentration Index of farming.** Caption credit: Elaborated from secondary data extracted from [26].

forestry products in the country. The entire eastern region of the state has increased its agricultural production with this activity since the mid-2000s. This fact justifies the concentration identified by the ICCs for farming in this region of the State.

Equally recent activities have been gaining strength in the agricultural structure of the State of Mato Grosso do Sul, as is the case with beekeeping, which has also spread due to the growth of the activity of planted forests, so that associations and small producers began to produce honey within the forest planting areas of pine and eucalyptus.

Another activity with an increasing number of projects in the state is freshwater aquaculture, which has grown especially in the last 10 years, and now the state has some refrigeration units accounting for almost 30.2% of the volume of slaughtered fish raised in the tanks in Brazil, according to [32]. The municipalities of Aparecida do Taboado with 40.97%, Selvíria with 32.99%, Brasilândia with 6.21%, Mundo Novo with 5.68%, and Itaporã with 4.21% are alone responsible for 90% of the production of captive tilapia of the State of Mato Grosso do Sul. In the past three years, a program of the government of the State of Mato Grosso do Sul called Propeixe has been seeking to foster the raising of fish and has goals to double the state's production by next year, according to [33].

Farming establishments have a distinct characteristic in relation to other sectors when analyzed by their size. Micro and small businesses add up to 55% of agricultural establishments while medium and large enterprises together add up to 45% of the total establishments in the State. Further, about 40% of medium and large sized establishments are primarily located in only ten municipalities: Campo Grande, Ribas do Rio Pardo, Três Lagoas, Maracaju, Água Clara, Dourados, Ponta Porã, Rio Brilhante, Corumbá, and Chapadão do Sul. Of the medium and large sized establishments, 51% are in the production of beef cattle, the other half in other crops and other services related to farming. Of the micro and small establishments, only 23% are destined to produce beef cattle and 35% for land preparation, cultivation, and harvesting services, the remaining 42% are destined for other crops or dairy cattle, in addition to activities supporting farming.

## Final considerations

The Bioceanic Corridor (CB), Bioceanic Route (RB), or *Rota de Integração Latino Americana* (RILA) is a transport corridor in the process of being implemented, which has led to recurring discussions. Certainly, this project will bring positive externalities and also challenges. Studies

applied in this context should thus guide public policies that can enhance the benefits from this route and minimize the negative impacts.

Based on the economic analyses conducted, it was possible to raise some important questions. It is likely that the Bioceanic Route does not result in links with the productive sector in several municipalities, configuring itself as a mere way route. The municipalities that may have the greatest benefit for being part of the route are majorly those in the farming sector. The fact that the large number of municipalities with little significance suggests a certain diversification may have as a background the fragility of these economies that are often dependent on public funds and the number of jobs and aid that the municipal or state government generates in the municipality. Thus, the deepening of agglomerations in the municipalities of the Bioceanic Route depends not only on the implementation or the logistical benefits but also on the decisions of allocation of companies, the size, the productive characteristic of each region that can interfere with the predictability.

The agglomerations verified coincide with some of the LPAs previously mapped by the Observatory, but in this analysis, we do not consider all the characteristics for the determination of an LPA, which is one of the reasons for differences. Based on the LPAs identified by the Observatory, important LPAs in the vicinity of the Route were identified, such as those related to the sectors of Beekeeping, Agro-industrial Horticulture, Medicinal and Herbal Plants, Manioc Crops, Forest Base, Milk, and Fish Farming. All of these are linked to primary production. Therefore, these are the sectors that should explore and seek to benefit from the new international road corridor to deepen or seek new markets, especially in Asia. For local productive tourism arrangements, it is essential to establish public policies that can contribute to the strengthening of the attraction of demand in neighboring countries, such as Paraguay, Argentina, and Chile, and to create tourist infrastructure.

Thus, this article identified and characterized the municipalities of Mato Grosso do Sul inserted in the context of this new international road corridor. In addition, it analyzed the productive structures through indicators. Finally, it analyzed the indicators of productive specialization. Thus, some of the main remarks in this first paper were 1) the mapping of the route, 2) the spatial identification and analysis of the productive structure of the municipalities of Mato Grosso do Sul participating in the Bioceanic Corridor designed, 3) the analysis of the characteristics of the productive specialization of the selected municipalities through indicators of concentration, and 4) the mapping of local productive arrangements in Mato Grosso do Sul.

The results reveal a window of opportunity for the public and private sector to enhance the opportunities that will come from this corridor. Public policies can enable integration towards competitiveness gains for these sectors. Mere unplanned integration will potentially only lead to the strengthening of regional inequalities. To put it more pessimistically, without conducting an elaborate and efficient planning and without an articulated public action, there might be several threats from the integration that may be greater than the opportunities offered. Another key issue for further studies concerns the environmental impacts and carbon emissions of this type of project, as evidenced by [34].

In this context, it is important to implement well-delineated public policies that foster greater competitiveness for the state sectors through the expansion of innovative potential of companies and the training of the workforce, promoting trade agreements, especially within MERCOSUR, in order to avoid integration problems (predatory competitions, tax wars, and the evasion of industrial parks), and encouraging complementarities and expansion of integrations of regional production chains in the search for increased added value and generation of higher-paying jobs. This policy would permit positive externalities observed in the economies of agglomeration. In this sense, it is particularly important to promote the consolidation of a regional innovation system, as pointed out by [35], that is, to promote cooperation between

the productive sector, companies, universities, and research institutes in search of products and services with a higher degree of novelty protected through patents. Thus, it would be possible at first, to reduce costs and to raise competitiveness, then later to add value and raise employment and income rates.

## Author Contributions

**Conceptualization:** Mateus Boldrine Abrita.

**Formal analysis:** Mateus Boldrine Abrita, Angelo Rondina Neto.

**Funding acquisition:** Ruberval Franco Maciel.

**Methodology:** Rafaella Stradiotto Vignandi, Daniel Amorim Souza Centurião.

**Project administration:** Nelagley Marques, Vanessa Aparecida de Moraes Weber.

**Supervision:** Mateus Boldrine Abrita.

**Validation:** Ana Paula Camilo Pereira.

**Visualization:** Ana Paula Camilo Pereira, Guilherme Espindola Junior.

**Writing – original draft:** Vanessa Aparecida de Moraes Weber.

**Writing – review & editing:** Vanessa Aparecida de Moraes Weber.

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
