## [Decision Letter · Decision Letter 0]

17 Jan 2023

PONE-D-22-29881Dynamics of local productive arrangements in the municipalities of Mato Grosso do Sul considering the transformations of the Bioceanic CorridorPLOS ONE

Dear Dr. Abrita,

Thank you for submitting your manuscript to PLOS ONE. After careful consideration, we feel that it has merit but does not fully meet PLOS ONE’s publication criteria as it currently stands. Therefore, we invite you to submit a revised version of the manuscript that addresses the points raised during the review process.

We look forward to receiving your revised manuscript.

Kind regards,

Wei-Haur Lam, PhD (QUB) FEI FIEI

Guest Editor

PLOS ONE

Journal Requirements:

5. We note that Figures 1 to 10 in your submission contain [map/satellite] images which may be copyrighted. All PLOS content is published under the Creative Commons Attribution License (CC BY 4.0), which means that the manuscript, images, and Supporting Information files will be freely available online, and any third party is permitted to access, download, copy, distribute, and use these materials in any way, even commercially, with proper attribution. For these reasons, we cannot publish previously copyrighted maps or satellite images created using proprietary data, such as Google software (Google Maps, Street View, and Earth). For more information, see our copyright guidelines: http://journals.plos.org/plosone/s/licenses-and-copyright.

  a. You may seek permission from the original copyright holder of Figures 1 to 10 to publish the content specifically under the CC BY 4.0 license.  

Additional Editor Comments :

Authors are invited to review the manuscript.

Reviewers' comments:

Reviewer's Responses to Questions

**Comments to the Author**

1. Is the manuscript technically sound, and do the data support the conclusions?

Reviewer #1: Yes

Reviewer #2: Yes

2. Has the statistical analysis been performed appropriately and rigorously? 

Reviewer #1: N/A

Reviewer #2: Yes

3. Have the authors made all data underlying the findings in their manuscript fully available?

Reviewer #1: Yes

Reviewer #2: Yes

4. Is the manuscript presented in an intelligible fashion and written in standard English?

Reviewer #1: Yes

Reviewer #2: Yes

5. Review Comments to the Author

Reviewer #1: Abstract - Result, research outcome and contribution should be addressed. Abstract should be revised.

Introduction - Suggest to rewrite as it is identical to abstract.

Line 66-67 - Repetition, with intro and abstract.

Content, methodology, data analysis and final results are well presented with very minor flaw. Suggest to make a final proofread as typo found.

Reviewer #2: The paper discussed the dynamics of local productive arrangements in the municipalities of Mato Grosso do Sul.The topic of this study is interesting.

(1)The abstract can be further refined. For example, the research results in the abstract can be summarized into three points.

(2)It is suggested to further improve the innovation points.

（3）The overall quality of English is good, but need to be checked carefully again. I suggest the authors should look for an English native speaker to further check the language of the paper.

（4）Some fresh paper should be added, like:

Sun, H., Samuel, C.A, Amissah, JCK, Taghizadeh-Hesary, F., Mensah, IA., 2020. Non-linear nexus between CO2 emissions and economic growth: A comparison of OECD and B&R countries, Energy, 212, 118637. https://doi.org/10.1016/j.energy.2020.118637.

6. PLOS authors have the option to publish the peer review history of their article (what does this mean?). If published, this will include your full peer review and any attached files.

Reviewer #1: No

Reviewer #2: No

---

## [Author Response · Author response to Decision Letter 0]

14 Feb 2023

Reviewer #1: Abstract - Result, research outcome and contribution should be addressed. Abstract should be revised.

Introduction - Suggest to rewrite as it is identical to abstract.

Line 66-67 - Repetition, with intro and abstract.

Content, methodology, data analysis and final results are well presented with very minor flaw. Suggest to make a final proofread as typo found.

The abstract was elaborated again taking into account the suggestions. The introduction and repetition of line 66-67 has been rewritten. A typing review was performed.

Reviewer #2: The paper discussed the dynamics of local productive arrangements in the municipalities of Mato Grosso do Sul.The topic of this study is interesting.

(1)The abstract can be further refined. For example, the research results in the abstract can be summarized into three points.

(2)It is suggested to further improve the innovation points.

（3）The overall quality of English is good, but need to be checked carefully again. I suggest the authors should look for an English native speaker to further check the language of the paper.

（4）Some fresh paper should be added, like:

Sun, H., Samuel, C.A, Amissah, JCK, Taghizadeh-Hesary, F., Mensah, IA., 2020. Non-linear nexus between CO2 emissions and economic growth: A comparison of OECD and B&R countries, Energy, 212, 118637. https://doi.org/10.1016/j.energy.2020.118637.

The abstract was redone and highlighted the innovations and results. In the introduction, emphasis was placed on research innovations. The research was revised by a specialized company with native speakers, as can be seen in the certificate included to the resubmission. The research suggestion was included.

---

## [Decision Letter · Decision Letter 1]

21 Mar 2023

Dynamics of local productive arrangements in the municipalities of Mato Grosso do Sul considering the transformations of the Bioceanic Corridor

PONE-D-22-29881R1

Dear Dr. Abrita,

We’re pleased to inform you that your manuscript has been judged scientifically suitable for publication and will be formally accepted for publication once it meets all outstanding technical requirements.

Kind regards,

Wei-Haur Lam, PhD (QUB) FEI FIEI

Guest Editor

PLOS ONE

Additional Editor Comments (optional):

Reviewers' comments:

Reviewer's Responses to Questions

**Comments to the Author**

1. If the authors have adequately addressed your comments raised in a previous round of review and you feel that this manuscript is now acceptable for publication, you may indicate that here to bypass the “Comments to the Author” section, enter your conflict of interest statement in the “Confidential to Editor” section, and submit your "Accept" recommendation.

Reviewer #1: All comments have been addressed

Reviewer #2: All comments have been addressed

2. Is the manuscript technically sound, and do the data support the conclusions?

Reviewer #1: Yes

Reviewer #2: Yes

3. Has the statistical analysis been performed appropriately and rigorously? 

Reviewer #1: Yes

Reviewer #2: Yes

4. Have the authors made all data underlying the findings in their manuscript fully available?

Reviewer #1: Yes

Reviewer #2: Yes

5. Is the manuscript presented in an intelligible fashion and written in standard English?

Reviewer #1: Yes

Reviewer #2: Yes

6. Review Comments to the Author

Reviewer #1: (No Response)

Reviewer #2: The spatial econometric methodology was adopted to determine the State's productive concentration, which is fine topic for me. Now after revisions, all my points are addressed already. So I think it already meet the acceptance standard of Plos One.

7. PLOS authors have the option to publish the peer review history of their article (what does this mean?). If published, this will include your full peer review and any attached files.

Reviewer #1: No

Reviewer #2: No

---

## [Editor Report · Acceptance letter]

27 Mar 2023

PONE-D-22-29881R1 

Dynamics of local productive arrangements in the municipalities of Mato Grosso do Sul considering the transformations of the Bioceanic Corridor 

Dear Dr. Abrita:

I'm pleased to inform you that your manuscript has been deemed suitable for publication in PLOS ONE. Congratulations! Your manuscript is now with our production department. 

Kind regards, 

on behalf of

Professor Wei-Haur Lam 

Guest Editor

PLOS ONE